# CGRP causes anxiety via HP1γ–KLF11–MAOB pathway and dopamine in the dorsal hippocampus
Narumi Hashikawa-Hobara ✉, Kyoshiro Fujiwara & Naoya Hashikawa

Calcitonin gene-related peptide (CGRP) is a neuropeptide that causes anxiety behavior; however, the underlying mechanisms remain unclear. We found that CGRP modulates anxiety behavior by epigenetically regulating the HP1γ-KLF-11-MAOB pathway and depleting dopamine in the dorsal hippocampus. Intracerebroventricular administration of CGRP (0.5 nmol) elicited anxiety-like behaviors in open field, hole-board, and plus-maze tests. Additionally, we observed an increase in monoamine oxidase B (MAOB) levels and a concurrent decrease in dopamine levels in the dorsal hippocampus of mice following CGRP administration. Moreover, CGRP increased abundance the transcriptional regulator of MAOB, Krüppel-like factor 11 (KLF11), and increased levels of phosphorylated heterochromatin protein (p-HP1γ), which is involved in gene silencing, by methylating histone H3 in the dorsal hippocampus. Chromatin immunoprecipitation assay showed that HP1γ was recruited to the *Klf11* enhancer by CGRP. Furthermore, infusion of CGRP (1 nmol) into the dorsal hippocampus significantly increased MAOB expression as well as anxiety-like behaviors, which were suppressed by the pharmacological inhibition or knockdown of MAOB. Together, these findings suggest that CGRP reduces dopamine levels and induces anxiety-like behavior through epigenetic regulation in the dorsal hippocampus.

Calcitonin gene-related peptide (CGRP) is a neuropeptide involved in various physiological processes. It is primarily expressed in sensory neurons, and has been implicated in the regulation of pain, inflammation, vasodilation, and migraine headaches[1–3]. CGRP-containing neurons are widely distributed throughout the central nervous system, with particularly high levels of expression in the hypothalamus, preoptic area, amygdala, thalamus, and hippocampus[4,5]. CGRP has been suggested to play a role in the modulation of emotional responses, including fear, anxiety, and depression[6–8]. We previously reported that CGRP is involved in depressive-like behavior[7] and fear memory[8,9], and in the present study, we focused on its role in anxiety. The mechanisms by which CGRP modulates anxiety responses still remain unclear.

Dopamine is a neurotransmitter that plays a role in a variety of physiological processes, including motor control, reward, and mood regulation. Dopamine has been implicated in the regulation of fear and anxiety responses in several brain regions, including the hippocampus[10,11]. Recent studies have begun to shed light into the role of dopamine receptors in anxiety. For example, it has been reported that lack of dopamine D2 receptors causes anxiety and depression[12], and that, in contrast, overexpression of these receptors in the dorsal raphe nucleus improves anxiety-like behavior[13]. Despite the abundant evidence of the involvement of D2 receptors in anxiety, the links between dopamine, CGRP and anxiety are still unclear. Although it was recently reported that intracerebroventricular (ICV) administration of CGRP antibody suppresses the upregulation of dopamine D2 receptors caused by infraorbital nerve ligation[14], neither changes in the hippocampus nor an association with anxiety were reported.

Here, we investigated the effects of CGRP, administered ICV or stereotaxic injection, on dopamine levels and anxiety-like behaviors in mice. We found that CGRP decreased dopamine, upregulated monoamine oxidase B (MAOB) and the transcriptional regulator of MAOB, Krüppel-like factor 11 (KLF11), and increased levels of phosphorylated heterochromatin protein (p-HP1γ), which is involved in gene silencing, by methylating histone H3 in the mouse hippocampus. Furthermore, stereotaxic injection of CGRP into the hippocampus induced anxiety-like behavior, and this effect was blocked by selegiline, a MAOB inhibitor or *MaoB* knockdown, suggesting a link between CGRP signaling and MAOB in the hippocampus.

Department of Life Science, Okayama University of Science, 1-1 Ridai-cho, Kita-ku, Okayama 700-0005, Japan.
✉e-mail: hashikawa-hobara@ous.ac.jp

Our findings suggest that CGRP reduces dopamine levels and induces anxiety-like behavior through epigenetic regulation in the dorsal hippocampus.

## Results

### Intracerebroventricular administration of CGRP produces anxiogenic effects, with lower levels of dopamine in the hippocampus

To evaluate the anxiogenic effects of CGRP, we injected CGRP into 8-week-old C57BL6J mice, and 24 h later, they were subjected to behavioral tests (Fig. 1a). In the open field test, locomotor activity was not affected by CGRP administration (Fig. 1b, c), but time spent in the central area was significantly decreased (Fig. 1d, p = 0.0124). Next, we performed the hole-board test, which can also detect anxiolytic behavior. CGRP did not affect the latency to the first head dip (Fig. 1e). However, CGRP significantly suppressed head dip behavior, indicating that it induces anxiety-like behavior (Fig. 1f, p = 0.0241). In the elevated plus maze test, CGRP-administered mice showed significantly reduced open arm entries (Fig. 1g, h, p < 0.0001) and time in the open arms (Fig. 1i, p = 0.0064), compared with saline treatment. Next, we examined whether CGRP (ICV) affects dopamine levels in the hippocampus. The hippocampus plays a critical role in the regulation of anxiety, and in particular, the activation of granule cells in the dentate gyrus suppresses anxiety behavior[15]. After behavioral paradigm, we collected the hippocampus tissues from mice and measured dopamine level by ELISA. CGRP administration significantly decreased dopamine levels in the hippocampus (Fig. 1j, p = 0.0076). Together, these results suggest that CGRP administration in the hippocampus decreases dopamine levels and exerts an anxiogenic effect.

### CGRP (ICV) increases abundance MAOB in the hippocampus

Because CGRP administration decreased hippocampal dopamine levels, we next examined whether CGRP affects dopamine-related factors. q-RT-PCR was used to measure *Th, Slc6a3, Ddc, Comt, Dbh, MaoA* and *MaoB* mRNA levels in the hippocampus (Fig. 2a). The data were analyzed using two-way ANOVA for statistical testing, followed by post hoc testing with Fisher's LSD (Fig. 2b). Notably, a significant increase in *MaoB* mRNA levels upon CGRP administration was observed (Fig. 2b, p = 0.0059), promoting further investigation into its protein expression. CGRP administration significantly increased MAOB protein expression in the hippocampus (Fig. 2c, p = 0.0359, Supplementary Fig. 1). These results suggest that CGRP increases MAOB expression and decreases dopamine levels in the mouse hippocampus.

### CGRP (ICV) increases abundance KLF11 by epigenetically modulating p-HP1γ

Because CGRP increased abundance MAOB and reduced dopamine levels, we next focused on MAOB transcriptional factors. Because several reports demonstrated that the transcription factor KLF11 activates *MaoB* gene expression[16,17], we examined whether CGRP affects KLF11expression in the hippocampus. qRT-PCR and western blot analysis revealed that KLF11 levels were significantly increased by CGRP (ICV) (Fig. 3a, p = 0.0243 and Fig. 3b, p = 0.0018, Supplementary Fig. 2).

Protein kinase A (PKA) is required for signal transduction downstream of CGRP[18]. Furthermore, a recent report showed that activation of PKA leads to phosphorylation of HP1γ, causing it to associate with euchromatin and activate gene transcription[19]. Thus, we examined whether CGRP impacts p-HP1γ levels. Compared with saline treatment, CGRP significantly increased p-HP1γ (Ser83) levels (Fig. 3c, p = 0.0177, Supplementary Fig. 2). Next, we performed chromatin immunoprecipitation (ChIP) assays to evaluate whether HP1γ binds to the *Klf11* enhancer or promoter region 24 h after CGRP administration in the hippocampus. The results were expressed as a percentage of relative binding. This was calculated by comparing the ChIP assay signal (bound HP-1γ) to the input sample signal (sample without anti-HP-1γ). CGRP treatment significantly decreased the binding of HP1γ to the *Klf11* enhancer site (Fig. 3d; −200 bp,

p = 0.001, −1000 bp, p = 0.0101 and −1500 bp, p = 0.0294), but not other sites (− 500 bp and −700 bp), compared with saline treatment. We also speculated that histone H3 methylation might be altered in CGRP-administered mice because of the increased p-HP1γ expression. The results were expressed as a percentage of relative binding. This was calculated by comparing the ChIP assay signal (bound H3K9me3) to the input sample signal (sample bound histone H3). CGRP significantly decreased the levels of methylated histone H3 bound to the *Klf11* enhancer site (Fig. 3e; −1000 bp, p = 0.041), but not other sites (− 200 bp, −500 bp, −700 bp and −1500 bp). These results suggest that CGRP increases HP1γ phosphorylation and decreases histone H3 methylation, thereby increasing *Klf11* transcription, which, in turn, increases MAOB expression. Furthermore, we conducted a ChIP assay to determine if KLF11 binds to the promoter site of mouse *MaoB*. Following the findings of Ou et al[16]., which reported KLF11 binding to the CACCC element in the human *MaoB* promoter, we designed primers targeting the promoter region of mouse *MaoB* (−279 bp to −163 bp) and performed the ChIP assay. The results were expressed as a percentage of relative binding. This was calculated by comparing the ChIP assay signal (bound KLF11) to the input sample signal (sample without anti-KLF11). All signals were normalized against the levels detected in the saline-treated sample, with the saline-treated level set to 100%. The results indicated significant binding of KLF11 to the mouse *MaoB* promoter site in response to CGRP (ICV) treatment (Fig. 3f, p = 0.0447).

### Stereotaxic intrahippocampal CGRP injection alters MAOB expression, decreases dopamine, and elicits anxiety-like behavior

Because we found that CGRP (ICV) increases abundance MAOB in the hippocampus, we examined whether CGRP-induced anxiety behavior involves the hippocampus. Mice receiving stereotaxic hippocampal CGRP or saline injection were subjected to behavioral testing. We also examined whether administration of the MAOB inhibitor selegiline (1 mg/kg, intraperitoneal administration for 3 times) suppresses the anxiety-like behavior induced by stereotaxic intrahippocampal CGRP injection (Fig. 4a). In the open field test, CGRP or selegiline injection did not affect locomotor activity (Fig. 4b, c). In contrast, for time spent in the central area, two-way ANOVA revealed a significant effect of selegiline (p = 0.005), but no selegiline × CGRP interaction, nor an effect of CGRP (Fig. 4b, d). In the hole-board test, there were no significant differences in the latency to the first head dip in the hole (Fig. 4e). For the head dip count, two-way ANOVA showed a significant CGRP × selegiline interaction (Fig. 4f, p = 0.0308). Tukey's *post hoc* analysis revealed significant differences between saline + saline and saline + CGRP treatments (p = 0.0145) as well as between the saline + CGRP and selegiline + CGRP treatments (p = 0.0344, Fig. 4f). We also conducted the elevated plus maze test (Fig. 4g). Two-way ANOVA revealed no significant differences in open arm entries (Fig. 4h). For time spent in the open arm, two-way ANOVA showed a significant effect of the CGRP × selegiline interaction (p = 0.0203), but no effect of CGRP or selegiline (Fig. 4i). Tukey's *post hoc* analysis revealed significant differences between saline + saline and saline + CGRP treatments (Fig. 4i, p = 0.0352). Together, these findings demonstrate that stereotaxic intrahippocampal CGRP injection has anxiogenic effects in the hole-board and plus maze tests. Subsequently, we investigated the expression of MAOB in the hippocampus. Our results indicated that CGRP injection significantly elevated MAOB expression in the hippocampus (p = 0.0383), an effect that was diminished by selegiline (Fig. 4j, p = 0.0084, Supplementary Fig. 3). Furthermore, CGRP injection led to a decrease in dopamine levels, while selegiline administration resulted in a significant increase (Fig. 4k, p = 0.0038). Our findings suggest that systemic selegiline administration mitigates CGRP-induced anxiety-like behavior in the hole board test. However, the intraperitoneal delivery of selegiline necessitates consideration of potential non-specific effects in areas beyond the intended target. To ascertain the necessity of hippocampal MAOB in CGRP-mediated anxiogenesis, we investigated the impact of *MaoB* knockdown on anxiety-like behaviors in mice. To confirm the impact of MaoB knockdown on MAOB expression in the mouse hippocampus, we quantified MAOB

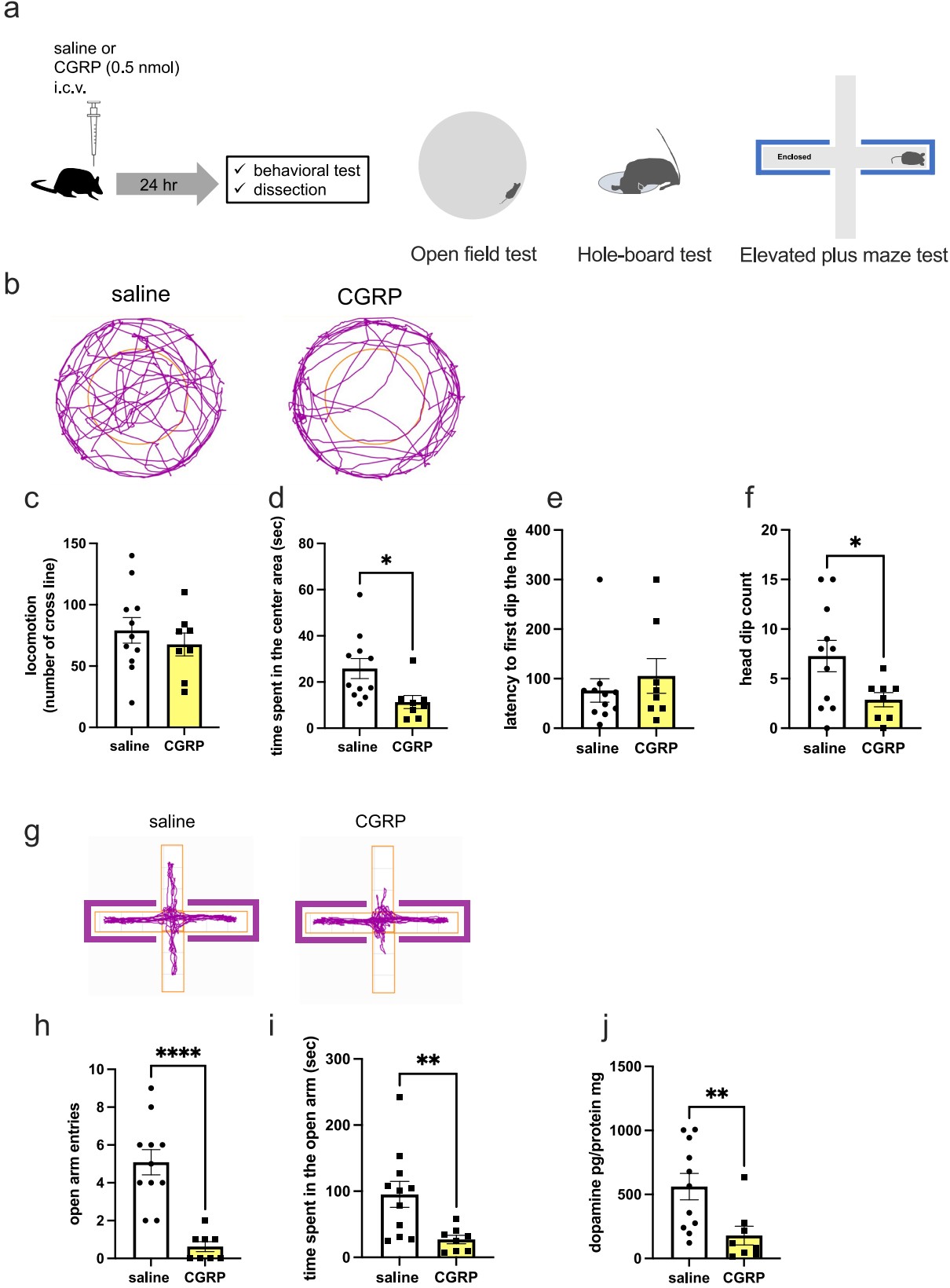

**Fig. 1 | Effects of CGRP administration on anxious behavior. a** Trial schematic for CGRP intracerebroventricular (ICV) administration (0.5 nmol) and anxious behavior tests. **b** Representative track plot in the open field test, (**c**) traveled distance, (**d**) time spent in the center area. Hole-board test showing (**e**) latency to first dip in the hole and (**f**) head dip count. **g** Representative track plot in the elevated plus maze test, (**h**) number of open arm entries and (**i**) time spent in the open arm. **j** Dopamine levels (pg/mg protein) in the hippocampus after behavioral testing 24 h after administration of CGRP. Each bar indicates the mean ± SEM, with significant differences shown as inserts. * $p < 0.05$, **$p < 0.01$, ****$p < 0.0001$. $n = 11$ saline and $n = 8$ CGRP. Welch's $t$ test.

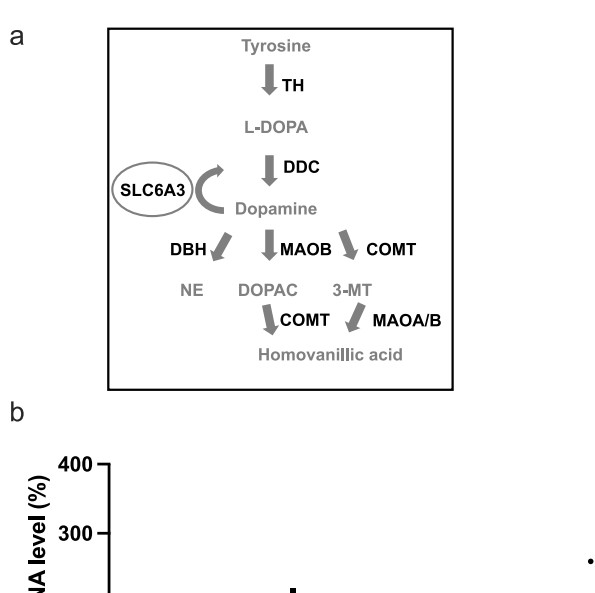

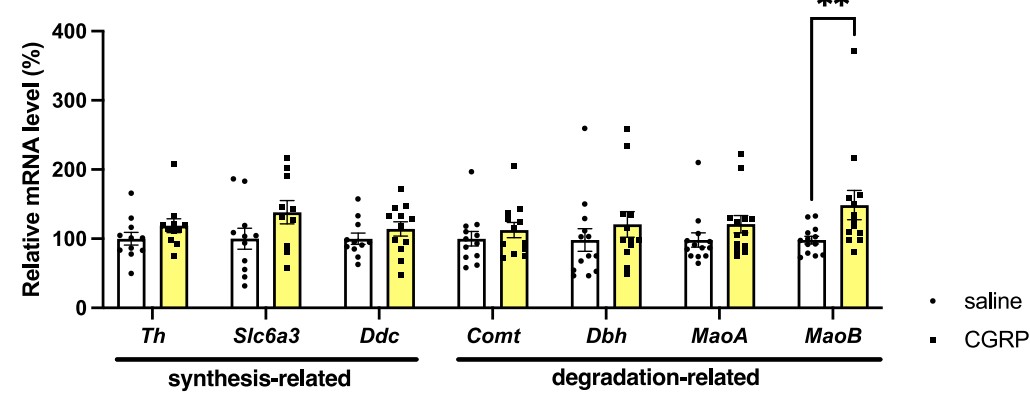

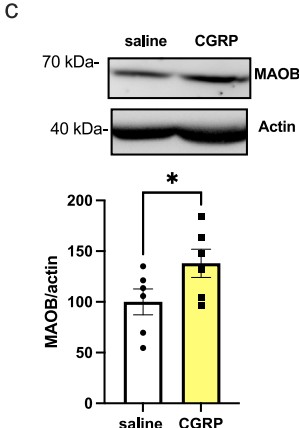

**Fig. 2 | Changes in dopamine metabolic enzyme in the hippocampus induced by CGRP (ICV). a** Schematic of the metabolic pathway of dopamine synthesis and clearance. **b** Tyrosine hydroxylase (*Th*) (*n* = 11 saline and *n* = 11 CGRP), dopamine transporter (*Slc6a3*) (*n* = 11 saline and *n* = 10 CGRP), dopa decarboxylase (*Ddc*) (*n* = 11 saline and *n* = 12 CGRP), catechol-O-methyltransferase (*Comt*) (*n* = 12 saline and *n* = 12 CGRP), dopamine beta-hydroxylase (*Dbh*) (*n* = 13 saline and *n* = 12 CGRP), monoamine oxidase A (*Maoa*) (*n* = 13 saline and *n* = 13 CGRP), monoamine oxidase B (*MaoB*) (*n* = 13 saline and *n* = 13 CGRP), and (**c**) MAOB protein (*n* = 6 saline and *n* = 6 CGRP). Each bar indicates the mean ± S.E.M. *$p < 0.05$. Two-way ANOVA multiple comparisons with Fisher's LSD (**b**). Welch's t test (**c**).

protein levels 4 days post-administration of *MaoB*-siRNA. This treatment resulted in a significant reduction of MAOB levels, as depicted in Fig. 5a ($p = 0.0387$, Supplementary Fig. 4). To further investigate MAOB's role in CGRP-mediated anxiogenesis, we administered either *MaoB*-siRNA or a non-targeting control (NTG) into the brains of mice on Day 1. Two days after the initial injection (Day 3), CGRP was administered directly to the hippocampus, followed by behavioral assessments the next day (Day 4) (Fig. 5b). In the open field test, *MaoB*-siRNA injection significantly reduced locomotor activity (Fig. 5c, d, $p = 0.0334$), whereas no significant difference was observed in time spent in the central area between NTG control and *MaoB*-siRNA groups (Fig. 5c, e). In the hole-board test, *MaoB*-siRNA notably decreased the latency to the first head dip (Fig. 5f, $p = 0.0418$), but

did not significantly affect the head dip count (Fig. 5g). In the elevated plus maze test, *MaoB*-siRNA-treated mice exhibited significantly more open arm entries (Fig. 5h, i, $p = 0.032$), but no change was observed in the time spent in the open arms (Fig. 5j). Collectively, these results imply that CGRP induces anxiety-like behavior by reducing dopamine levels through the upregulation of MAOB expression in the dorsal hippocampus of mice.

## Discussion

Here, we showed that injecting CGRP into the hippocampus significantly increases abundance MAOB and decreases hippocampal dopamine, thereby inducing anxiety-like behavior. Furthermore, this effect is mediated by the epigenetic regulation of heterochromatin phosphorylation. Figure 6

**Fig. 3 | CGRP phosphorylates HP1γ and activates KLF11 transcription in the hippocampus.**
**a** Krüppel-like factor 11 (*Klf11*) mRNA ($n = 7$ saline and $n = 5$ CGRP), (**b**) KLF11 protein ($n = 8$ saline and $n = 7$ CGRP), (**c**) p-HP1γ expression ($n = 6$ saline and $n = 7$), and (**d**) chromatin, obtained from the mouse hippocampus, were immunoprecipitated using antibodies against HP1γ ( − 200 bp; $n = 14$ saline and $n = 16$ CGRP, −500 bp; $n = 15$ saline and $n = 8$ CGRP, −700 bp; $n = 7$ saline and $n = 15$ CGRP, −1000 bp; $n = 14$ saline and $n = 16$, −1500 bp; $n = 14$ saline and $n = 15$ CGRP), and (**e**) methylation of histone H3K9 ( − 200 bp; $n = 13$ saline and $n = 13$ CGRP, −500 bp; $n = 8$ saline and $n = 8$ CGRP, −700 bp; $n = 13$ saline and $n = 13$ CGRP, −1000 bp; $n = 13$ saline and $n = 13$ CGRP, −1500 bp; $n = 12$ saline and $n = 13$ CGRP).
**f** Chromatin, obtained from the mouse hippocampus after saline or CGRP ICV, was immunoprecipitated using KLF11 antibody against KLF11 ($n = 9$ saline and $n = 7$ CGRP). Each bar indicates the mean ± S.E.M. *$p < 0.05$, **$p < 0.01$, Welch's t test (**a–c, f**). Two-way ANOVA multiple comparisons with Fisher's LSD (**d, e**).

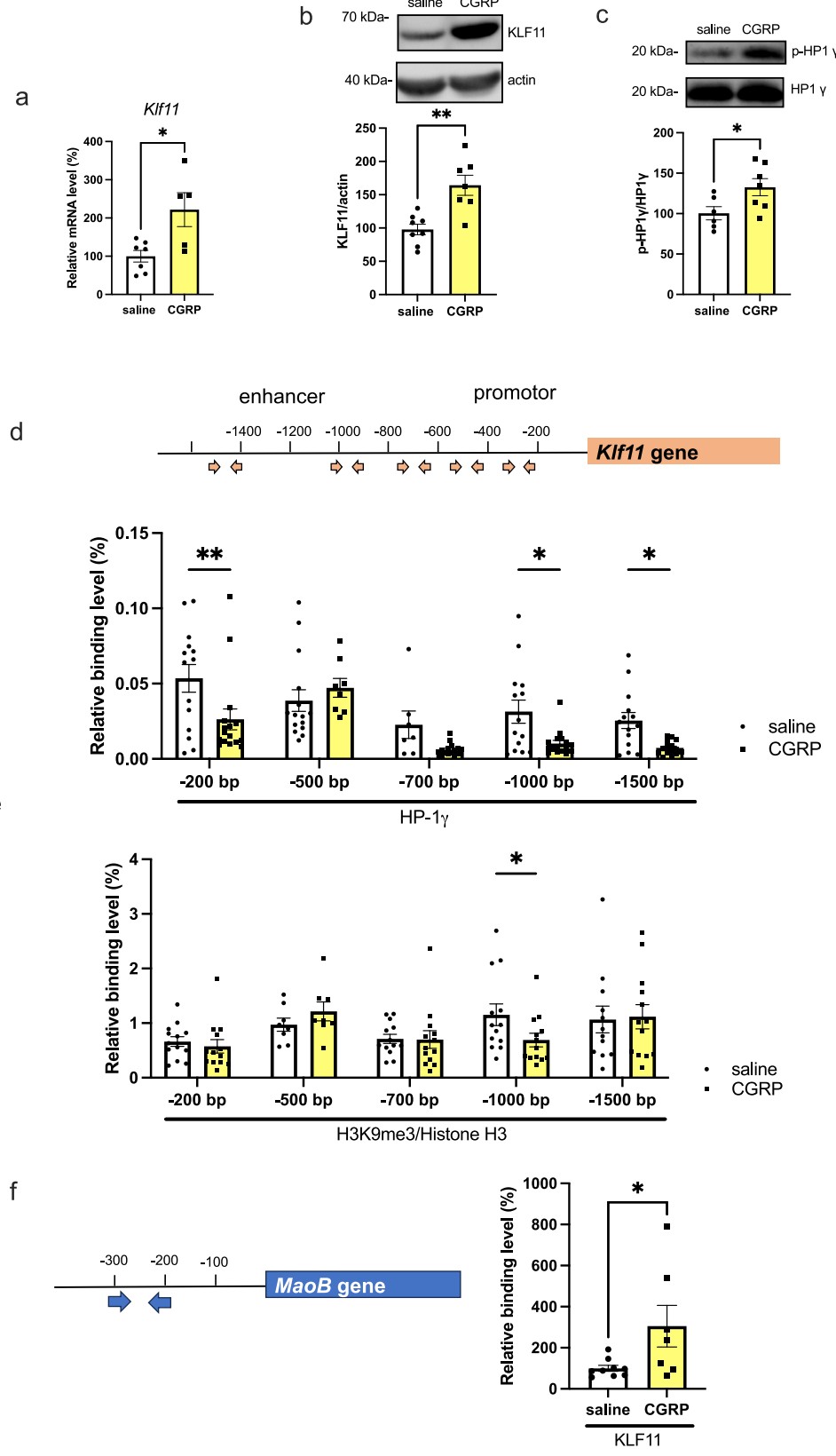

illustrates the molecular mechanisms underlying the anxiogenic action of CGRP based on our current findings. In a stressful situation or amphetamine exposure, dopamine release can promote anxiety-like behavior[20,21]. Similarly, genetic or pharmacological manipulations that increase dopamine signaling elicit anxiety-like behavior[22,23]. Conversely, decreased dopamine release in the prefrontal cortex, which is involved in cognitive and emotional processing, has been linked to increased anxiety-like behavior[24]. Thus, dopamine release can have both anxiogenic and anxiolytic effects, depending on the brain region. Previous studies show that deficiency of dopamine D2 receptors causes anxiety and depressive symptoms[13], and that intrahippocampal sulpiride (a dopamine D2 receptor blocker) injection induces anxiety-like behavior[25]. In accordance with these previous

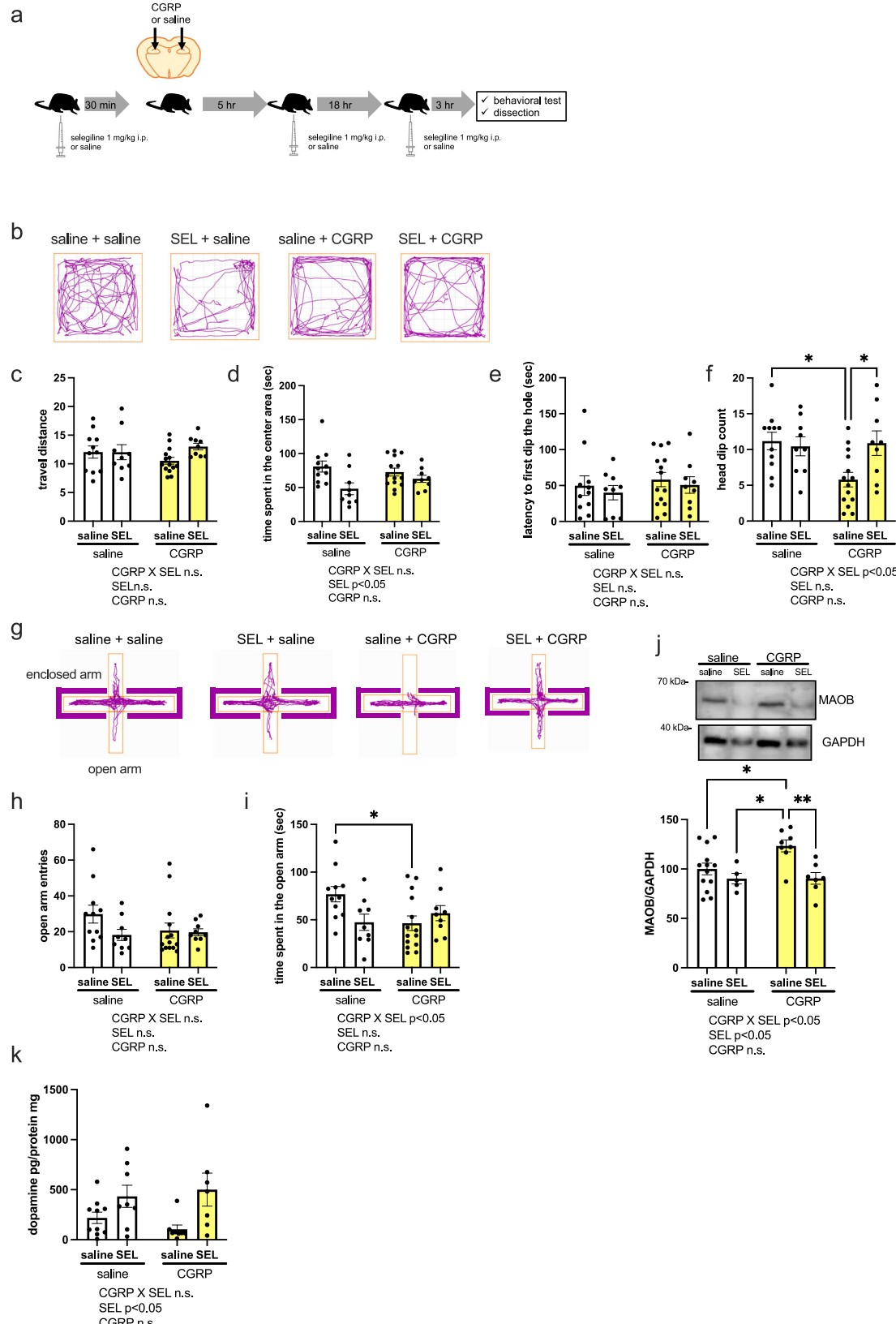

**Fig. 4 | Stereotaxic intrahippocampal CGRP injection induces anxiety-like behavior that is suppressed by selegiline. a** Schematic of CGRP and selegiline injections and anxiety behavioral tests ($n = 11$ saline + saline, $n = 9$ saline + selegiline, $n = 14$ CGRP + saline, $n = 9$ CGRP + selegiline). **b** Representative track plot in the open field test, (**c**) traveled distance and (**d**) time spent in the center area. Hole-board test showing (**e**) latency to first head dip in the hole and (**f**) head dip count. **g** Representative track plot in the elevated plus maze test, (**h**) number of open arm entries and (**i**) time spent in the open arm. **j** MAOB protein expression ($n = 13$ saline + saline, $n = 5$ saline + selegiline, $n = 8$ CGRP + saline, $n = 7$ CGRP + selegiline). **k** Dopamine levels (pg/protein mg) in the hippocampus ($n = 10$ saline + saline, $n = 8$ saline + selegiline, $n = 8$ CGRP + saline, $n = 7$ CGRP + selegiline). Each bar indicates the mean ± S.E.M. *$p < 0.05$, **$p < 0.01$. Two-way ANOVA multiple comparisons with Tukey's post hoc test. If a statistical interaction was observed between factors, comparison of all four groups was performed by Tukey's post hoc test.

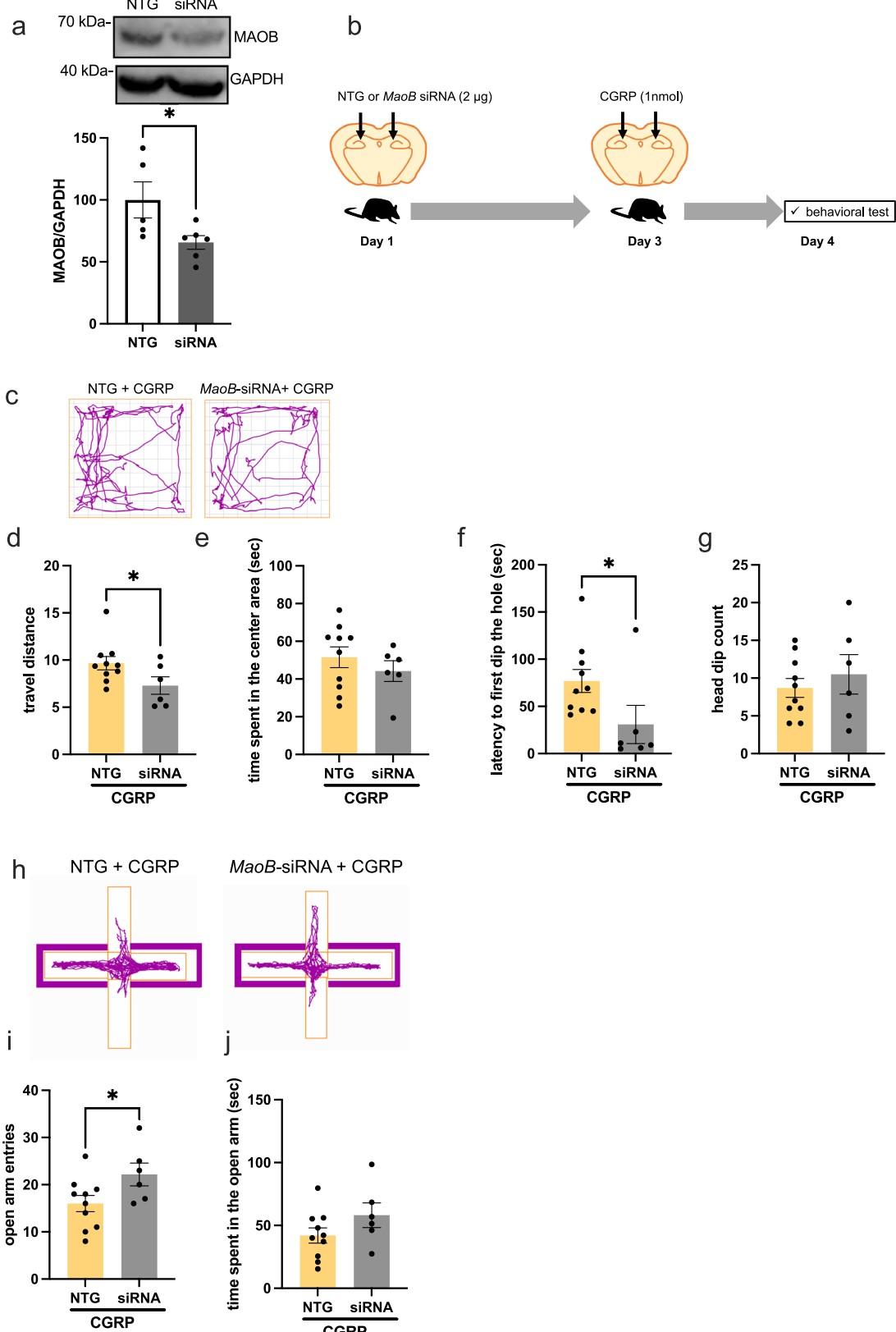

**Fig. 5 | CGRP-mediated anxiogenic is required for the dorsal hippocampal MAOB expression. a** MAOB protein expression 4 days following the administration of MAOB siRNA ($n = 5$ NTG, $n = 6$ siRNA). **b** Schematic of CGRP and *MaoB*-siRNA injections and anxiety behavioral tests ($n = 10$ NTG + CGRP, $n = 6$ *MaoB*-siRNA + CGRP). **c** Representative track plot in the open field test, (**d**) traveled distance and, (**e**) time spent in the center area. Hole-board test showing (**f**) latency to first dip in the hole and (**g**) head dip count. **h** Representative track plot in the elevated plus maze test, (**i**) number of open arm entries and (**j**) time spent in the open arm. Each bar indicates the mean $n \pm$ SEM, with significant differences shown as inserts. * $p < 0.05$. Welch's t test.

observations, our present results show that CGRP (ICV) induces anxiety-like behavior that is associated with reduced hippocampal dopamine levels. This finding is supported by intra-dorsal hippocampal injection of CGRP, which produced anxiogenic effects associated with dopamine decrease. In rodents, studies show that damage or genetically induced dysfunction of the hippocampus leads to increased anxiety-like behavior[26,27]. It has been suggested that the various hippocampal regions subserve distinct functions, with the dorsal hippocampus involved in memory, learning and spatial learning, while the ventral hippocampus cooperates with the amygdala to regulate anxiety-related behavior[28–30]. More recently, it was shown that the dorsal CA1 subregion of the hippocampus is activated by an anxiogenic environment[31] and that genetic manipulation of the dorsal hippocampus has an anxiogenic effect[32,33]. Thus, the dorsal hippocampus plays a critical role in the regulation of anxiety behavior.

Intraventricular administration and stereotaxic intrahippocampal CGRP injection differed in their behavioral effects, including time spent in the central area in the open field test, and open arm entries in the elevated plus maze test. It is not uncommon to obtain different behavioral test results when comparing different administration routes, such as intraventricular and hippocampal administration. We also used different apparatuses for the open field test for the ICV and hippocampal administration routes, making it more difficult to compare the results. In a previous study, a 48-h period of water deprivation in the rat induced anxiety-like behaviors in the open field and elevated plus maze tests, and changed gene expression in the lateral habenula, and basolateral and central amygdala[34]. Thus, the lack of anxiety-like symptoms in the open field test following intrahippocampal CGRP injection suggests that another brain region may be involved in the anxiolytic effect of CGRP administered ICV. However, we observed anxiety responses in the elevated plus maze and hole-board tests following intra-hippocampal CGRP injection. Furthermore, we also observed changes in MAOB expression, suggesting that increased dopamine catabolism is involved in the induction of anxiety-like behavior.

Selegiline, also known as L-deprenyl, is a medication primarily used to treat symptoms associated with Parkinson's disease, as well as major depressive disorder in some cases. In our study, we observed that treatment selegiline alone exhibited an anxiogenic effect, evidenced by a reduction in the time spent in the open arms of the plus maze test and in the center area of the open filed test. One hypothesis is that intraperitoneal administration of selegiline might influence other brain regions. Additionally, administering selegiline three times before the behavioral testing could have amplified other effects. Notably, 'anxiety' is listed as a side effect in the selegiline drug package insert[35]. While our current understanding is limited, it is important to note that selegiline, a MAOB inhibitor, appears to suppress CGRP-mediated anxiety response in mice. There are three closely related HP1 isoforms in mammals—HP1α, β and γ. These proteins bind histone methylated at H3K9, leading to gene silencing and heterochromatin formation, and have critical roles in cell cycle regulation, DNA repair and RNA splicing[36–38]. Although all three HP1 proteins are localized in heterochromatin, HP1γ, which is phosphorylated at Ser83, is present in euchromatin[39]. Phosphorylation of Ser83 appears to impair the silencing activity of HP1γ[40] and positively participates in splicing[41]. In the present study, we found that CGRP (ICV) increased p-HP1γ levels, leading to reduction of its binding to methylated histone (H3K9) at the *Klf11* enhancer site, in turn increasing KLF11 expression. These results suggest that CGRP attenuates gene silencing activity by enhancing HP1γ phosphorylation. Thus, CGRP(ICV) increases MAOB expression through epigenetic regulation of the PKA–HP1γ–KLF11 pathway. However, it remains unclear whether the CGRP-mediated increase in p-HP1γ is directly involved in the methylation of histone H3K9. Histone H3K9 undergoes various methylation processes, including monomethylation (H3K9me), dimethylation (H3K9me2), and trimethylation (H3K9me3), mediated by histone methyltransferases (HMTases). Suppressor of variegation 3-9 homologue1 (SUV39H1) and SUV39H2 are key mammalian HMTases[42]. Notably, SUV39H1 has a preferential affinity for H3K9me1, suggesting that H3K9me1 is essential for the enzymatic activity of SUV39H1[43].

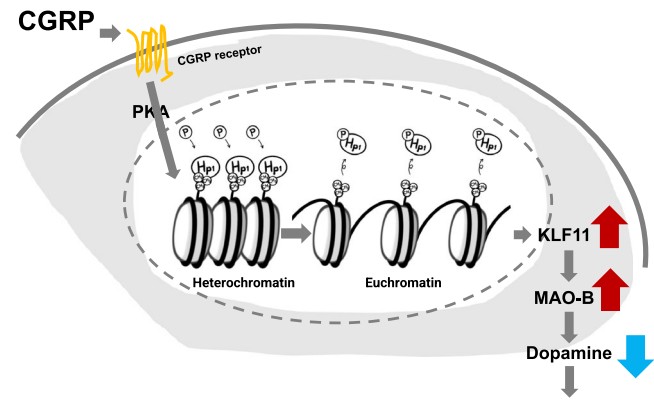

**Fig. 6 | Model of CGRP-induced anxiety-like behavior through epigenetic regulation of the p-HP1γ–KLF11–MAOB–dopamine pathway.** CGRP promotes the phosphorylation of HP1γ, leading to the detachment of HP1γ from methylated histone H3, thereby loosening chromatin condensation and activating the transcription of KLF11. KLF11, in turn, increases the transcription of the dopamine metabolizing enzyme MAOB, suggesting that the activation of dopamine metabolism induces anxiety.

Additionally, the catalytic activities of SUV39H1/H2 are augmented upon binding to H3K9me2 and H3K9me3[44]. In mammalian systems, the recruitment of SUV39H1/H2 is further facilitated by the binding of HP1α and β to H3K9me2 and H3K9me3[45–47]. Another report demonstrated that phosphorylation of Ser83-HP1γ by PKA activation results in its localization to euchromatin, by immunofluorescence staining of H3K9me3[19]. Additionally, HP1γ elicits the methylation of histone H4K20 in human cancer tissue and H3K36 in embryonic stem cells[37,48]. Although these previous observations seem to support our present findings, several outstanding questions remain. These include investigating whether HP1γ is responsible for recruiting H3K9me3, assessing how CGRP related SUV39H1/H2 activity, and elucidating its subsequent effects in neuronal cells. Addressing these issues is crucial for advancing our understanding and warrants further detailed investigation.

Although further study is needed to clarify the mechanisms of mood-modulating activities of CGRP, the peptide may have potential therapeutic applications for anxiety and other psychiatric disorders.

## Methods
### Animals
C57BL/6 J male mice were obtained from Shimizu Laboratory Supplies Co., Ltd. (Kyoto, Japan) and habituated for 2 weeks. Animals at 9–10 weeks of age were used in the experiments. All animals were housed in the Animal Research Center of Okayama University of Science, at a controlled ambient temperature of 22 °C, 50 ± 10% relative humidity, and a 12/12-h light/dark cycle (lights on at 8:00 A.M.). Mice were housed in groups of four or five per cage (23.5 × 35.3 × 16.0 cm, width × length × height) with paper roll for environmental enrichment, and food and water were available *ad libitum*. All behavioral experiments were performed between 10:00 A.M. and 4:00 P.M. Animal procedures were approved by the Okayama University of Science Animal Care and Use Committee (authorization numbers 2021-002). We have complied with all relevant ethical regulations for animal use. In accordance with these guidelines, efforts were made to minimize the number of animals used and their suffering. All behavioral paradigm were conducted under white illuminance.

### Behavioral assessments

**Open field test**. The open field test was performed as previously described[7,49]. In the ICV administration experiment, 11 mice were administered saline, and eight mice were administered CGRP. A circular open field was used (57.5 cm in diameter, 32 cm in height, with the floor

**Table 1 | Oligonucleotide sequences for real-time PCR amplification**

|        | Forward | Reverse |
|--------|---------|---------|
| *Th*     | CAGCTGGAGGATGTGTCTCA | GAAAATCACGGGCAGACAGT |
| *Slc6a3* | CCTGTGGAAGGGAGTAAAGACTTCAG | GTAGAAGTCCACACTGAGGTATGCTC |
| *Ddc*    | CTCAGGATTCATCACTGACTACAGGC | GACTCAAACTCATGAGACAGCTCCAC |
| *Comt*   | CTGGAGCTAGGAGCTTATTGTGG | CAGCGTAGTCAGGGTTAATCTCC |
| *Dbh*    | CCTCTCAGCTTCATACACACCTG | CTGTAGTGGTTGTCCCTGTTCAC |
| *MaoA*   | GCAGCTAGAGAGGTCTTGAATGC | GTTCCTCTCTAAGAAGGTGTGGG |
| *MaoB*   | ATTAGTGCCATTCCACCTGC | AACTGAACCCAAAGGCACAC |
| *Klf11*  | CCCACTGACAAAGGTCAACAGAC | GGATCAGGGACAGAAATCAGAGG |
| *Actin*  | GGTCAGAAGGACTCCTATGTG | GGTGTGGTGCCAGATCTTCTCC |

divided into 19 sections)[7]. All animal behaviors were recorded using a digital camera. Line crossings and the time spent in the central area were measured using a stopwatch and a counter by a blinded investigator. For behavioral testing in mice given stereotaxic intrahippocampal CGRP injection, we used a square open field (36.5 × 36.5 × 33 cm, width × length × height)[49], and 11 mice were administered saline, 9 mice were administered saline and selegiline, 13 mice were administered CGRP and saline, and 9 mice were administered CGRP and selegiline. The white light intensity was regulated at approximately 40 LUX. Mice were placed in the center of the open field chamber and tested for 3 min. The total distance traveled and time (s) spent in the center of the open field were analyzed using Any-Maze behavior tracking software (Muromachi Kikai Co., Ltd., Tokyo, Japan).

**Hole-board test**. The hole-board test was performed as described previously[50], with modification. In this test, there is a negative correlation between head-dipping activity and anxiety state[51]. Briefly, the apparatus consisted of a box (50 × 50 × 40 cm height) with four equally spaced holes, 3 cm in diameter, in the floor. The behavior of each mouse was monitored using a video camera (JVC KENWOOD Corp., Tokyo, Japan). Each animal was placed in the center of the hole-board and allowed to freely explore the apparatus for 5 min. The white light intensity was regulated at approximately 300 LUX. We measured the latency to the first head dip and the number of head dips.

**Elevated plus maze test**. The elevated plus maze test was conducted as previously described[52], with some modification. The apparatus consisted of two open arms (15 × 5 cm) and two enclosed arms of the same size with 15-cm-high transparent walls, and the arms were connected by a central square (5 × 5 cm). The arms were elevated 35 cm above the floor. Each mouse was placed in the central square of the maze, facing one of the open arms and allowed to move freely for 5 min. The white light intensity was regulated at approximately 150 LUX. The time spent and the number of entries in the open arms were measured with Any-Maze behavior tracking software.

**ELISA**
Dopamine levels were measured in dorsal hippocampal homogenates by ELISA according to the manufacturer's instructions (BioVision, Milpitas, CA). Briefly, Rinse the tissues with ice-cold PBS and homogenized in 100 µL PBS with a homogenizer on ice. Then sonicate the suspension with an ultrasonic cell disrupter. The homogenates are then centrifuged for 5 min at 5,000 g to retrieve the supernatant. Protein concentrations were measured using DC protein assay kits (Bio-Rad Laboratories, Inc., Tokyo, Japan).

**Quantitative real-time PCR (qRT-PCR) analysis**
Animals were sacrificed by the administration of an overdose of pentobarbital-Na (100 mg/kg). Total RNA was extracted from the dorsal hippocampus, placed in RNAlater (Life Technologies Co., Tokyo, Japan), and stored at –30 °C. Total RNA extraction and qRT-PCR were performed as previously described[8]. Primer sequences, designed by the authors, used for qRT-PCR are listed in Table 1. The threshold cycle values for the target genes (tyrosine hydroxylase (*Th*)), solute carrier family 6 (*Slc6a3*) (a dopamine transporter), dopa decarboxylase (*Ddc*), catechol-o-methyltransferase (*Comt*), dopamine beta-hydroxylase (*Dbh*), monoamine oxidase A (*MaoA*), monoamine oxidase B (*MaoB*), Krüppel-like factor 11 (*Klf11*) and the internal control gene (*Actin*) were determined. *Actin* was demonstrating the most stable cycle threshold values, was chosen as the house keeping gene.

**Western blotting**
Western blot analysis was performed as previously described[9]. Briefly, dorsal hippocampal samples were resolved by SDS-PAGE and transferred to polyvinylidene difluoride membranes (HybondP, GE Healthcare UK Ltd.). The membrane was blocked with a blocking agent (GE Healthcare) and then incubated at 4 °C overnight with the following primary and secondary antibodies: mouse monoclonal anti-MAOB (1:1,000; Santa Cruz Biotechnology, Inc., sc-515354), mouse monoclonal anti- TIEG2 (KLF11) (1:1,000; Santa Cruz Biotechnology, sc-136101), rabbit polyclonal anti-p-HP1γ (Ser83) (1:5,000; Invitrogen, Thermo Scientific, PA517210), mouse monoclonal anti-HP1γ (1:5,000; Santa Cruz Biotechnology, sc-398562), mouse monoclonal anti-actin (1:10,000; Santa Cruz Biotechnology, sc-8432), horseradish peroxidase-conjugated secondary antibody against mouse or rabbit (1:20,000; Santa Cruz Biotechnology, sc-2357 for mouse, sc-2004 for rabbit). The antibody-reactive bands were visualized using a chemiluminescent substrate kit (GE Healthcare). Bands were analyzed by densitometry, using ImageJ (https://imagej.nih.gov/ij/).

**Drug treatments**
**ICV administration**. ICV administration was performed as previously described[8]. Briefly, rat CGRP (0.5 nmol, injected volume, 5 µL, PEPTIDE Institute, Inc., Osaka, Japan) was diluted in saline. Isoflurane (1.5%–2.0%) was used for brief anesthesia during the injections. Hole-board and EPM tests were performed 24 h after ICV administration of CGRP. The EPM test was monitored with a video camera (JVC KENWOOD), and the time spent and number of entries in the open and closed arms were recorded.

**Stereotaxic surgery**. Mice were anesthetized with a mixture of three anesthetic agents administered intraperitoneally as previously described[53]. Selegiline hydrochloride (1 mg/kg; Kyowa Pharmaceutical Industry Co., Ltd., Osaka, Japan) was administered intraperitoneally 30 min before stereotactic injection of CGRP. CGRP was injected with a glass pipette bilaterally into the dorsal hippocampus, using the following coordinates from the Bregma: anteroposterior: 3 mm, mediolateral: 2 mm, dorsoventral: 2 mm (flow rate, 5 nL/s; injected volume, 0.5 µL).

**Table 2 | Oligonucleotide sequences for ChIP**

|  | Forward | Reverse |
|---|---|---|
| Klf11 −1500 bp | CAAAGGCTCCCTCTCCAAAT | CCATCCAACGTCCCTACTGT |
| Klf11 −1000 bp | TGCTGCACTGTAGGTTGGAG | GGCCTTTAGGTGCGGTTATT |
| Klf11 −700 bp | GTGGGTGTCCTTTGTATGGC | TAAGCTGCCGTCTGCTACAA |
| Klf11 −500 bp | GCGGCAGAAGAGGACCTTAC | CGTGAAGCCTGGAAAGTAGG |
| Klf11 −200 bp | CCATTGGCCCGCTTCTTG | GCAAACCAAAATACACCGCT |
| MaoB | GACGGACTTTCAGGTTCCAG | GGGTGGAGCTCTTAACCCTC |

Mice were administered selegiline 5 h after surgery and 3 h before behavioral testing on the next day.

**Chromatin immunoprecipitation (ChIP)**

A total of 35 mice were used in this experiment. ChIP was performed as previously described[8]. Briefly, CGRP or saline was administered ICV 24 h before cross-linking. The mice were deeply anesthetized using a combination of three anesthetic agents: medetomidine hydrochloride (Domitol, Meiji Seika Pharma Co., Ltd., Tokyo, Japan, at 0.3 mg/kg), midazolam (Dormicum, Astellas Pharma Inc., Tokyo, Japan, at 4.0 mg/ kg), and butorphanol (Vetorphale, Meiji Seika Pharma Co., Ltd., at 5.0 mg/kg), all administered intraperitoneally. They were then perfused transcardially with saline, followed by 1% paraformaldehyde at pH 7.4. The hippocampus was post-fixed for 10 minutes in 1% paraformalde- hyde, after which 330 mM glycine was added. Subsequently, the chro- matin was sheared into fragments of approximately 0.5–1 kb and immunoprecipitated using anti-HP1γ antibody (1:250 for the hippo- campal sample; Santa Cruz Biotechnology), anti-Histone H3 (trimethyl K9) antibody (1:100; Abcam, ab8898), anti-Histone H3 antibody (1:100; Abcam, ab10799), or TIEG2 (KLF11) antibody (1:100 Santa Cruz Bio- technology, sc-136101). For immunoprecipitated DNA fragments were used for qPCR (Eco Real-Time PCR System (Illumina Inc.)), using PCR primers specific for Klf11 or MaoB, which were designed around the putative promoter regions of Klf11 or MaoB (Table 2).

**siRNA constructs**. The short interfering RNA (siRNA) reagents used were Dharmacon's Accell siRNA, SMARTpool (Accell Mouse MaoB [109731] siRNA-SMART pool, 10 nmol) and Non-targeting siRNA (GE Healthcare). Non-targeting control or MaoB-siRNA were injected with a glass pipette bilaterally into the dorsal hippocampus, using the following coordinates from the Bregma: anteroposterior: 3 mm, mediolateral: 2 mm, dorsoventral: 2 mm (flow rate, 5 nL/s; injected volume, 0.5 μL). Additionally, CGRP was injected into the dorsal hippocampus on day3, followed by the assessment of anxiety-like behavior on day4.

**Statistics and reproducibility**

All data are expressed as the mean ± S.E.M. GraphPad Prism 9 software (GraphPad Software Inc., San Diego, CA, USA) was used for all statistical analyses. Data were assessed by investigators blinded to group assignment. Comparisons between two values were analyzed using Welch's $t$ test. Two- way analysis of variance (ANOVA) multiple comparisons with Fisher's LSD (Fig. 2b, Fig. 3d, e). Two-way ANOVA was also performed when comparing four values (Fig. 4). If there was a significant difference in the interaction between groups (CGRP and selegiline), Tukey's post hoc test was used to compare all groups. For A $P$ value of <0.05 was considered statistically significant.

**Reporting summary**

Further information on research design is available in the Nature Portfolio Reporting Summary linked to this article.

**Data availability**

Numerical source data for figures and plots can be found in supplemen- tary data 1.

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

## Acknowledgements

KAKEN supported this work (grant number 21K07532). We thank Barry Patel, PhD, from Edanz (https://jp.edanz.com/ac), for editing a draft of this manuscript.

## Author contributions

N.H. and N.H.-H. designed the experiments; N.H., K.F. and N.H.-H. conducted the experiments; N.H.-H. wrote the main manuscript text; K.F. and N.H-H. conducted the statistical analyses and prepared the figures. All authors reviewed and approved the manuscript.

## Competing interests

The authors declare no competing interests.
