## [Peer Review File · Communications Biology]

Reviewers' comments:

Reviewer #1 (Remarks to the Author):

In their article Hashikawa-Hobara and colleagues describe a set of experiments to investigate the underlying mechanisms that lead to an induction of anxiety behavior by the Calcitonin gene-related peptide. The topic is relevant and the study well designed. A set of experiments builds on each other to detangle the exact mechanisms. Unfortunately, statistical results are not reported in a standardized way. Exact p values and effect sizes are missing. Therefore, it is hard to fully interpret the results. In addition, a few minor revisions could help to improve the manuscript overall.

1. In the abstract there is an unnecessary repetition of the report on increased MAOB and decreased dopamine.
2. The introduction ends with a summary of the results, which should be omitted.
3. On Page 7 in the results section there is speculation of the mechanisms 'by increasing MAOB expression via the epigenetic regulation of HP1 γ '. As this is a speculation based on results after ICV injection, it should be moved to the discussion but not be mentioned in the result section.
4. The discussion seems to oversell in some parts. As the effect of CGPR on epigenetic regulation has only been investigated after ICV injection authors should be careful to state that the effect after hippocampal injection is mediated by epigenetic effects; it might be mediated. In addition, the increase of KLF11 after CGRP administration does not necessarily prove that KLF11 is involved in the anti-inflammatory effect of CGRP.
5. Data on selegiline are not discussed. I understand that they are hard to interpret (also given that the anxiogenic effect after hippocampal injection was not as high as after ICV injection). Can authors speculate on the anxiogenic effect of selegiline alone (in open field and EPM, saline group).
6. What was the rationale for using different open field mazes?
7. How was qRT-PCR analyzed? Are the results in the graphs CT values as percent of Actin CT values? Was it sufficient to use a single housekeeping gene

Reviewer #2 (Remarks to the Author):

In this manuscript, Hashikawa-Hobara and colleagues aim to dissect the signaling mechanisms downstream of CGRP action in hippocampus in the context of anxiety, following up on work from the same group reporting that CGRP in hippocampus modulates depressive-like and fear-related behaviors. The manuscript main novel findings – which are well summarized in the helpful Fig. 5 schematic – are that pro-anxiogenic effects of CGRP depend on the activation of a signaling cascade involving – in subsequent steps – HP1g phosphorylation and H3 histone methylation at the promoter of the KLF11 transcription factor, KLF11-mediated transcriptional regulation of monoamine oxidase MAOB, ultimately leading to change in hippocampal dopamine levels, and increased anxiety.

Overall, these conclusions, although mainly descriptive, are relatively well supported by the data, extend current literature and might be of interest to the general readership of Communications Biology. However, addressing a few weaknesses – some conceptual, some experimental/methodological and some in data visualization and manuscript overall clarity – could further increase the reliability of the data and the enthusiasm for the findings. Concerns are listed below in descending order of importance. Please note that only #1, #2 and maybe #4 would require some additional experimental work, while the rest can be addressed mostly with extended analyses and statistics or in text.

1. The major conceptual weakness in this study is the unconvincing selegiline (SEL) data in Fig. 4. Authors argue, before that last experiment, that increased MAOB expression and decreased dopamine levels in dorsal hippocampus are pro-anxiety. Inhibition of MAOB action with SEL by itself should then be expected to be anti-anxiety: yet, the inverse is observed in Fig. 4D,4J, with pro-anxiety effects at baseline in SEL-treated mice versus saline controls, in the absence of CGRP treatment. As SEL is administered systemically (ip), non-specific effects in other brain regions might understandably be responsible for such inconsistencies. It is nevertheless important, as the authors tried with this last experiment, to demonstrate some level of causality in the signaling cascade otherwise simply described in this manuscript – systemic SEL treatment simply fails to do so convincingly in the experiments presented in the current Fig. 4. Additional experiments should increase the levels of specificity in probing selected steps of the signaling cascade at play: one option, likely technically challenging, would be to administer SEL locally in dorsal hippocampus with canulae and mini-pumps. Another, simpler, alternative would be to manipulate (knock-down, likely) MAOB expression levels in hippocampus (or of other key players in that cascade) using viral strategies – of note, viral vectors for MAOB knock-down have been published (PMID: 36716674). See also KLF11 vectors (PMID: 32196819).
2. While the authors performed ChIP for transcriptional regulators on the KLF11 promoter, they did not confirm the next step, ie KLF11 binding on MAOB gene promoter by ChIP-qPCR. This would be required – with the current data, there is absolutely no indication that KLF11 does indeed regulate MAOB in this region, in this context, yet it is strongly suggested in the text and Fig.5 schematic. At the very least, is there a consensus KLF11 binding site in the mouse MAOB promoter?
3. The ChIP data in Fig. 3, as it is currently presented, is not up to current standards in the field. How are the “Relative binding levels” values obtained? Fig. 3 seems to normalize H3K9me3 over total H3 (and not H3K9, please correct in axis labels), is that the case? Have the authors used normalization versus inputs, or versus an IgG pulldown for HP1g ChIP? These data should be plotted on a y-axis scaled relative to

input (or IgG, or H3 at the very least), which should moreover be the same for each DNA locus. Authors might want to plot them on the same graph, with a single axes pair. Moreover, authors should definitely use 2way-ANOVA for omnibus statistical testing, and use post-hoc testing with correction for multiple comparisons at each DNA locus to identify where differences lie.

4. Another missing mechanistic link is between HP1g and H3K9me3 at the KLF11 promoter. Authors mention the histone methyltransferase SUV39H1 in the Discussion, but it might be worth trying to ChIP for it. Another approach (although without gene locus-specificity) might be to use IHC to assess co-localization of pHP1g, H3K9me3 and/or SUV39H1 in Hoechst-stained heterochromatin.

5. In Fig. 4K,4L, it is surprising to note that biochemical endpoints are only assessed after hippocampus-specific CGRP treatment without SEL treatment. Validating that SEL treatment (or any alternative causal experiment, see Comment #1) also affects dopamine levels, or other intermediate signaling endpoints if manipulating more upstream players, is important.

6. a. Are the tissue samples used in Fig. 2 and 3 from the animals in Fig. 1? If so, it would be interesting to try to correlate molecular endpoints (mRNA or protein expression) with behavioral performance to assess whether the molecular response to CGRP can predict behavioral response, especially in the absence of more causal experiments (see Comment #1). To avoid Simpson's paradox-related artefacts, correlations should be run independently in controls and CGRP-treated animals as well as together.

b. Similarly, authors should test for a correlation between pHPYg, KLF11 and MAOB protein levels, again especially in the absence of more causal experiments such as confirmed KLF11 binding onto MAOB promoter (see Comment #2).

7. The title and abstract should precise that dorsal, not ventral, hippocampus is targeted. Similarly throughout the main text, 'dorsal hippocampus' should be used instead of just 'hippocampus' for clarity.

8. For all legends, it is imperative for statistics to be reported with more detail. Always include t statistics, degrees of freedom (approximated if using Welch's correction) and exact p-values for all t-tests, and F values with numerator and denominator degrees of freedom for ANOVAs.

9. a. Fig. 2 B-H could benefit from being plotted as a single panel, and authors should consider applying multiple comparison corrections to their repeated t-tests.

b. A related point would be to distinguish in that panel between synthesis-related, pre-synaptic, axonally-expressed mRNAs (Th, Slc6a3, Ddc, Dbh) and degradation-related, likely in part post-synaptic, actually hippocampal mRNAs. Have authors confirmed (with IHC or RNA FISH) that Maob (in particular) is expressed post-synaptically by hippocampal cells? If so, what specific cell types in dorsal hippocampus? It is particularly important to be sure of this as other molecular experiments (ChIP, KLF11 and HP1g Western blots) are looking at hippocampal nuclear, hence post-synaptic, mechanisms.

10. Precise that ChIP experiments are looking at H3K9me3. Please note that this abbreviation is more standard (and more precise) that Met H3K9 as in Fig. 3. Please discuss me3 versus me2/me1 in Discussion.

11. In Fig. 1, please consider adding representative traces or heatmaps to quantification, similar to Fig. 4B,G.

12. In Fig. 1B,G and Fig. 4H,I, please simply plot one graph with the number of entries in open arms instead of two plots with total entries and % open arm entries. This is confusing. Keep plots of total time spent in open arms (Fig. 1H and Fig. 4J), of course.

13. In the Introduction, the lengthy discussion of similarities and differences between anxiety, depression

and fear could be removed or significantly shortened for improved flow and clarity.

14. Discussion of CGRP and KLF11 anti-inflammatory functions sound outside the direct relevance of this study. Please consider shortening or removing from Discussion.

15. In Fig. 5 schematic, please consider removing, or annotating with a question mark, the role of PKA which is not at all studied here.

16. In Methods, precise if behavioral testing is performed under red or white light, and add lux intensity.

17. In Methods, please detail tissue processing for ELISA.

18. In Methods, TIGE2 should be TIEG2 (KLF11).

19. In Methods, please give more details on ChIP protocol: tissue shearing parameters, fragment length, use of input or IgG controls (see Comment #3)

20. In Results, precise that SEL is administered ip (although alternative strategies should probably replace this experiment, see Comment #1)

21. Results text, Fig. 1I. Precise method used for dopamine levels quantification.

22. Results text, Fig. 2I. Precise this is MAOB protein expression, by contrast to mRNA levels above.

23. Introduction: "overexpression of these receptors in the dorsal raphe nucleus improves anxiety-like behavior". Add reference.

Reviewer #1 (Remarks to the Author):

1. In the abstract there is an unnecessary repetition of the report on increased MAOB and decreased dopamine.

Response: We thank the Reviewer for this comment. We revised the abstract to omit an unnecessary repetition of MAOB and dopamine description.

2. The introduction ends with a summary of the results, which should be omitted.

Response: We now omit the summary of the results in the introduction.

3. On Page 7 in the results section there is speculation of the mechanisms ‘by increasing MAOB expression via the epigenetic regulation of HP1γ’. As this is a speculation based on results after ICV injection, it should be moved to the discussion but not be mentioned in the result section.

Response: We appreciate the Reviewer’s suggestion. The last sentence of page 7 “Collectively, these findings suggest that CGRP induces anxiety-like behavior by decreasing dopamine levels by increasing MAOB expression via the epigenetic regulation of HP1γ. Figure 5 illustrates the molecular mechanisms underlying the anxiogenic action of CGRP based on our current findings.” was omitted and transported to the first discussion section.

page 8 L177-178.

“Figure 6 illustrates the molecular mechanisms underlying the anxiogenic action of CGRP based on our current findings.”

4. The discussion seems to oversell in some parts. As the effect of CGPR on epigenetic regulation has only been investigated after ICV injection authors should be careful to state that the effect after hippocampal injection is mediated by epigenetic effects; it might be mediated. In addition, the increase of KLF11 after CGRP administration does not necessarily prove that KLF11 is involved in the anti-inflammatory effect of CGRP.

Response: We appreciate the Reviewer’s comment. In the epigenetic discussion section, we mentioned that CGRP administration with ICV. **page 10 L219, L222.** We also appreciate the opportunity to clarify that anti-inflammatory effect of CGRP. According to the suggestion, we deleted the last sentence about anti-inflammatory effect.

5. Data on selegiline are not discussed. I understand that they are hard to interpret (also given that the anxiogenic effect after hippocampal injection was not as high as after ICV injection). Can authors speculate on the anxiogenic effect of selegiline alone (in open field and EPM, saline group).

Response: We concur with the reviewer on the importance of exploring why selegiline treatment alone exhibited anxiogenic effects in the open field test. One possibility is that intraperitoneal administration of selegiline might affect other brain regions. Additionally, administering it three times before the behavioral testing could have intensified other effects. Notably, 'anxiety' is listed as a side effect in the selegiline drug package insert (<https://www.drugs.com/sfx/selegiline-side-effects.html>). While our current understanding is limited, it's important to note that selegiline, a MAOB inhibitor, appears to suppress CGRP-mediated anxiety responses in mice.

6. What was the rationale for using different open field mazes?

Response: We have been conducting research on CGRP and behavior for an extended period. Initially, we used circular tanks for the open field tests, but we recently transitioned to square tanks and have been utilizing video analysis software in our experiments. According to the reviewer's comment, we re-examined the mouse track plot data and observed that in mice administered with CGRP, the avoidance of the center was distinctly captured in circular tanks as well as in square ones (Fig. 1B).

7. How was qRT-PCR analyzed? Are the results in the graphs CT values as percent of Actin CT values? Was it sufficient to use a single housekeeping gene

Response: In this experiment, Actin, GAPDH, and ribosomal RNA were selected as housekeeping genes. Ultimately, Actin, demonstrating the most stable CT (cycle threshold) values, was chosen as the housekeeping gene.

Reviewer #2 (Remarks to the Author):

1. The major conceptual weakness in this study is the unconvincing selegiline (SEL) data in Fig. 4. Authors argue, before that last experiment, that increased MAOB expression and decreased dopamine levels in dorsal hippocampus are pro-anxiety. Inhibition of MAOB action with SEL by itself should then be expected to be anti-anxiety: yet, the inverse is observed in Fig. 4D,4J, with pro-anxiety effects at baseline in SEL-treated mice versus saline controls, in the absence of CGRP treatment. As SEL is administered systemically (ip), non-specific effects in other brain regions might understandably be responsible for such inconsistencies. It is nevertheless important, as the authors tried with this last experiment, to demonstrate some level of causality in the signaling cascade otherwise simply described in this manuscript – systemic SEL treatment simply fails to do so convincingly in the experiments presented in the current Fig. 4. Additional experiments should increase the levels of specificity in probing selected steps of the signaling cascade at play: one option, likely technically challenging, would be to administer SEL locally in dorsal hippocampus with canulae and mini-pumps. Another, simpler, alternative would be to manipulate (knock-down, likely) MAOB expression levels in hippocampus (or of other key players in that cascade) using viral

strategies – of note, viral vectors for MAOB knock-down have been published (PMID: 36716674).

See also KLF11 vectors (PMID: 32196819).

Response: We acknowledge the importance of specifically inhibiting MAOB. In response to the reviewer's comment, and considering the technical challenges of minipump experiments as well as the unavailability of viral vectors, we opted for the *MaoB*-siRNA construct. The resulting data have been added as Figure 5. In the open field test, treatment with *MaoB*-siRNA significantly reduced the travel distances (D). In the hole board test, the siRNA treatment notably decreased the latency to first explore a hole (F). Additionally, in the plus maze test, there was a significant increase in entries into the open arms (I).

We also revised results section and method section as below.

Page 7, L150-L171.

“Our results indicated that CGRP injection significantly elevated MAOB expression in the hippocampus, an effect that was diminished by selegiline (Fig. 4J). Furthermore, CGRP injection led to a decrease in dopamine levels, while selegiline administration resulted in a significant increase (Fig. 4K). Our findings suggest that systemic selegiline administration mitigates CGRP-induced anxiety-like behavior in the hole board test. However, the intraperitoneal delivery of selegiline necessitates consideration of potential non-specific effects in areas beyond the intended target. To ascertain the necessity of hippocampal MAOB in CGRP-mediated angiogenesis, we investigated the impact of *MaoB* knockdown on anxiety-like behaviors in mice. We administered either *MaoB*-siRNA or a non-targeting control (NTG) into the brains of mice. Two days after the initial injection, CGRP was administered directly to the hippocampus, followed by behavioral assessments the next day (Fig. 5A). In the open field test, *MaoB*-siRNA injection significantly reduced locomotor activity (Fig. 5B, 5C), whereas no significant difference was observed in time spent in the central area between NTG control and *MaoB*-siRNA groups (Fig. 5B, 5D). In the hole-board test, *MaoB*-siRNA notably decreased the latency to the first head dip (Fig. 5E), but did not significantly affect the head dip count (Fig. 5F). In the elevated plus maze test, *MaoB*-siRNA-treated mice exhibited significantly more open arm entries (Fig. 5G, 5H), but no change was observed in the time spent in the open arms (Fig 5I). Additionally, MAOB protein levels in the hippocampus of the mice were measured. Collectively, these results imply that CGRP induces anxiety-like behavior by reducing dopamine levels through the upregulation of MAOB expression in the dorsal hippocampus of mice.”

Page 16 L367-374.

“siRNA constructs

The short interfering RNA (siRNA) reagents used were Dharmacon's Accell siRNA, SMARTpool (Accell Mouse *MaoB* [109731] siRNA-SMART pool, 10 nmol) and Non-targeting siRNA (GE Healthcare). Non-targeting control or *MaoB*-siRNA were injected with a glass pipette bilaterally into the dorsal hippocampus, using the following coordinates from the Bregma: anteroposterior: 3 mm, mediolateral: 2 mm, dorsoventral: 2 mm (flow rate, 5 nL/s; injected volume, 0.5 μ L). Additionally, CGRP was injected into the dorsal hippocampus on day3, followed by the assessment of anxiety-like behavior on day4.”

2. While the authors performed ChIP for transcriptional regulators on the KLF11 promoter, they did not confirm the next step, ie KLF11 binding on MAOB gene promoter by ChIP-qPCR. This would be required – with the current data, there is absolutely no indication that KLF11 does indeed regulate MAOB in this region, in this context, yet it is strongly suggested in the text and Fig.5 schematic. At the very least, is there a consensus KLF11 binding site in the mouse MAOB promoter?

Response: We appreciate the opportunity to provide additional information about our research strategy. According to the literature, the human *MaoB* promoter contains a KLF11 binding site, known as the CACCC element (Ou et al., 2004). Based on this report, we designed primers for the promoter region of mouse *MaoB* (spanning from -279 bp to -163 bp). Our results show that CGRP icv treatment significantly increased KLF11 binding at the *MaoB* promoter site, and we have included these findings as Figure 3F. We also revised results section as below.

Page 6, L119-127.

“Furthermore, we conducted a ChIP assay to determine if KLF11 binds to the promoter site of mouse *MaoB*. Following the findings of Ou et al¹⁶., which reported KLF11 binding to the CACCC element in the human *MaoB* promoter, we designed primers targeting the promoter region of mouse *MaoB* (-279 bp to -163 bp) and performed the ChIP assay. The results were expressed as a percentage of relative binding. This was calculated by comparing the ChIP assay signal (bound KLF11) to the input sample signal (sample without anti-KLF11). All signals were normalized against the levels detected in the saline-treated sample, with the saline-treated level set to 100%. The results indicated significant binding of KLF11 to the mouse *MaoB* promoter site in response to CGRP (ICV) treatment (Fig. 3F).”

3. The ChIP data in Fig. 3, as it is currently presented, is not up to current standards in the field. How are the “Relative binding levels” values obtained? Fig. 3 seems to normalize H3K9me3 over total H3 (and not H3K9, please correct in axis labels), is that the case? Have the authors used normalization versus inputs, or versus an IgG pulldown for HP1g ChIP? These data should be plotted on an y-axis scaled relative to input (or IgG, or H3 at the very least), which should moreover be the same for each DNA locus. Authors might want to plot them on the same graph, with a single axes pair. Moreover, authors should definitely use 2way-ANOVA for omnibus statistical testing, and use post-hoc testing with correction for multiple comparisons at each DNA locus to identify where differences lie.

Response: Thank you for your feedback. We have presented “Relative binding levels” as follows: The results were expressed as a percentage of relative binding. This was calculated by comparing the ChIP assay signal (bound HP-1γ) to the input sample signal (sample without anti-HP-1γ). For the histone methylation analysis, we compared the ChIP assay signal (bound H3K9me3) to the input sample signal (bound histone H3).

This explanation has been included in the ChIP results section.

Page 5 L106-109.

“The results were expressed as a percentage of relative binding. This was calculated by comparing the ChIP assay signal (bound HP-1 γ) to the input sample signal (sample without anti-HP-1 γ).”

Page 5 L112-114.

“The results were expressed as a percentage of relative binding. This was calculated by comparing the ChIP assay signal (bound H3K9me3) to the input sample signal (sample bound histone H3).”

Page 6 L122-126.

“The results were expressed as a percentage of relative binding. This was calculated by comparing the ChIP assay signal (bound KLF11) to the input sample signal (sample without anti-KLF11). All signals were normalized against the levels detected in the saline-treated sample, with the saline-treated level set to 100%.”

We also revised Figure 3 to plot data on the same graph, with a single axes pair. We did statistical analysis to use 2 way-ANOVA for omnibus statistical testing, and use post-hoc testing for multiple comparisons, Fisher’s LSD. According to the statistical analysis, CGRP treatment significantly decreased the binding of HP1 gamma to the KLF11 enhancer site (-200 bp, -1000 bp and -1500 bp) and the levels of methylated histone H3 bound to the KLF11 enhancer site (-1000 bp).

4. Another missing mechanistic link is between HP1 γ and H3K9me3 at the KLF11 promoter.

Authors mention the histone methyltransferase SUV39H1 in the Discussion, but it might be worth trying to ChIP for it. Another approach (although without gene locus-specificity) might be to use IHC to assess co-localization of pHP1 γ , H3K9me3 and/or SUV39H1 in Hoechst-stained heterochromatin.

Response: We concur that this is an important matter warranting further discussion. Currently, we are addressing SUV39H1 in the Discussion section (page 10, L221-225). Although there are numerous issues that still require consideration, we propose to move on to the next topic at this juncture.

page 11, L236-240.

“Although these previous observations seem to support our present findings, several outstanding questions remain. These include investigating whether HP1 γ is responsible for recruiting H3K9me3, assessing how CGRP related SUV39H1/H2 activity, and elucidating its subsequent effects in neuronal cells. Addressing these issues is crucial for advancing our understanding and warrants further detailed investigation.”

5. In Fig. 4K,4L, it is surprising to note that biochemical endpoints are only assessed after hippocampus-specific CGRP treatment without SEL treatment. Validating that SEL treatment (or

any alternative causal experiment, see Comment #1) also affects dopamine levels, or other intermediate signaling endpoints if manipulating more upstream players, is important.

Response: We are grateful for the opportunity to offer further details about our research approach. Additional data on MAOB and dopamine levels are presented in the Figure 4J and 4K. Meanwhile, we have re-examined the dopamine levels for Figure 1J.

6. a. Are the tissue samples used in Fig. 2 and 3 from the animals in Fig. 1? If so, it would be interesting to try to correlate molecular endpoints (mRNA or protein expression) with behavioral performance to assess whether the molecular response to CGRP can predict behavioral response, especially in the absence of more causal experiments (see Comment #1). To avoid Simpson's paradox-related artefacts, correlations should be run independently in controls and CGRP-treated animals as well as together.

b. Similarly, authors should test for a correlation between pHPYg, KLF11 and MAOB protein levels, again especially in the absence of more causal experiments such as confirmed KLF11 binding onto MAOB promoter (see Comment #2).

Response: Thank you for your constructive feedback. We would like to clarify that the data presented in Figures 2 and 3 and the animal model depicted in Figure 1 are different. This study was initially inspired by our prior research on fear memory, where we hypothesized a regulatory role for CGRP in dopamine modulation. This hypothesis led to the experiments conducted in the current study. Specifically, the data in Figure 2 were obtained from earlier experiments and, therefore, do not correspond to the animal model shown in Figure 1.

7. The title and abstract should precise that dorsal, not ventral, hippocampus is targeted. Similarly throughout the main text, 'dorsal hippocampus' should be used instead of just 'hippocampus' for clarity.

Response: We appreciate this recommendation. We now write out "dorsal hippocampus".

8. For all legends, it is imperative for statistics to be reported with more detail. Always include t statistics, degrees of freedom (approximated if using Welch's correction) and exact p-values for all t-tests, and F values with numerator and denominator degrees of freedom for ANOVAs.

Response: We appreciate this recommendation. We now add all p-values and F values in legends.

9. a. Fig. 2 B-H could benefit from being plotted as a single panel, and authors should consider applying multiple comparison corrections to their repeated t-tests.

Response: We appreciate the Reviewer's comment. We combined Fig. 2B to 2H into one figure as Fig. 2B. Furthermore, the presentation of data in Figure 2 has been revised and updated after reanalysis of the mRNA data. We also revised the main text as below.

Page 4 L87-L92.

"The data were analyzed using two-way ANOVA for statistical testing, followed by post-hoc testing with Fisher's LSD (Fig. 2B). Notably, a significant increase in *MaoB* mRNA levels upon CGRP administration was observed (Fig. 2B, $p = 0.0059$), promoting further investigation into its protein expression. CGRP administration significantly increased MAOB protein expression in the hippocampus (Fig. 2C)."

b. A related point would be to distinguish in that panel between synthesis-related, pre-synaptic, axonally-expressed mRNAs (*Th*, *Slc6a3*, *Ddc*, *Dbh*) and degradation-related, likely in part post-synaptic, actually hippocampal mRNAs. Have authors confirmed (with IHC or RNA FISH) that *Maob* (in particular) is expressed post-synaptically by hippocampal cells? If so, what specific cell types in dorsal hippocampus? It is particularly important to be sure of this as other molecular experiments (ChIP, KLF11 and HP1g Western blots) are looking at hippocampal nuclear, hence post-synaptic, mechanisms.

Response: We concur that distinguishing between synthesis-related and degradation-related aspects is intriguing. Accordingly, we have included descriptions in Fig. 2B as either 'synthesis-related' or 'degradation-related'. Previously, we stained the dorsal hippocampus with a MAOB antibody and, upon high magnification observation, identified MAOB-like immunoreactivity (indicated by white arrows) resembling nerve fibers. However, determining whether this immunoreactivity is pre-synaptic or post-synaptic was challenging. At present, we do not possess additional data on this matter. Nonetheless, we will certainly consider exploring this idea further.

10. Precise that ChIP experiments are looking at H3K9me3. Please note that this abbreviation is more standard (and more precise) that Met H3K9 as in Fig. 3. Please discuss me3 versus me2/me1 in Discussion.

Response: We appreciate the Reviewer's comment. We revised "Met H3K9/H3K9" to "H3K9me3/Histone H3". We also discuss me3 versus me2/me1 in Discussion section as below.

P10 L225- L240.

"Histone H3K9 undergoes various methylation processes, including monomethylation (H3K9me), dimethylation (H3K9me₂), and trimethylation (H3K9me₃), mediated by histone methyltransferases (HMTases). Suppressor of variegation 3-9 homologue1 (SUV39H1) and SUV39H2 are key mammalian HMTases⁴¹. Notably, SUV39H1 has a preferential affinity for H3K9me₁, suggesting that H3K9me₁ is essential for the enzymatic activity of SUV39H1⁴². Additionally, the catalytic activities of SUV39H1/H2 are augmented upon binding to H3K9me₂ and H3K9me₃⁴³. In mammalian systems, the recruitment of SUV39H1/H2 is further facilitated by the binding of HP1 α and β to H3K9me₂ and H3K9me₃^{44,45,46}. Another report demonstrated that phosphorylation of Ser83-HP1 γ by PKA activation results in its localization to euchromatin, by immunofluorescence staining of H3K9me₃¹⁹. Additionally, HP1 γ elicits the methylation of histone H4K20 in human cancer tissue and H3K36 in embryonic stem cells^{36,47}. Although these previous observations seem to support our present findings, several outstanding questions remain. These include investigating whether HP1 γ is responsible for recruiting H3K9me₃, assessing how CGRP

related SUV39H1/H2 activity, and elucidating its subsequent effects in neuronal cells. Addressing these issues is crucial for advancing our understanding and warrants further detailed investigation.”

11. In Fig. 1, please consider adding representative traces or heatmaps to quantification, similar to Fig. 4B,G.

Response: According to the reviewer’s comment, we added representative track plot both the open field test and plus maze test (Fig.1B and 1G).

12. In Fig. 1B,G and Fig. 4H,I, please simply plot one graph with the number of entries in open arms instead of two plots with total entries and % open arm entries. This is confusing. Keep plots of total time spent in open arms (Fig. 1H and Fig. 4J), of course.

Response: We appreciate the Reviewer’s comment. We added open arm entries graph instead of both total entries graph and open arm entries (%) in Fig. 1H and Fig. 4J.

13. In the Introduction, the lengthy discussion of similarities and differences between anxiety, depression and fear could be removed or significantly shortened for improved flow and clarity.

Response: We appreciate the reviewer's comment and have revised the introduction for greater clarity. Briefly, we have omitted descriptions from our previous report and removed the distinctions between anxiety, depression, and fear.

14. Discussion of CGRP and KLF11 anti-inflammatory functions sound outside the direct relevance of this study. Please consider shortening or removing from Discussion.

Response: We appreciate the Reviewers suggestion. We deleted the last sentence about anti-inflammatory effect.

15. In Fig. 5 schematic, please consider removing, or annotating with a question mark, the role of PKA which is not at all studied here.

Response: We appreciate the Reviewer’s comment. The explanation of PKA has been omitted.

16. In Methods, precise if behavioral testing is performed under red or white light, and add lux intensity.

Response: We appreciate the Reviewer’s comment. All behavioral paradigms were performed under white light of 40LUX for the open field test, 300LUX for hole board test and 150 LUX for the plus maze test. **Page 12 L257, L271, L282, L292.**

17. In Methods, please detail tissue processing for ELISA.

Response: According to reviewer's comment, we added detail tissue homogenates processing by the manufacturer's instructions.

Page 13 L298-301.

"Briefly, Rinse the tissues with ice-cold PBS and homogenized in 100 μ L PBS with a homogenizer on ice. Then sonicate the suspension with an ultrasonic cell disrupter. The homogenates are then centrifuged for 5 min at 5,000 g to retrieve the supernatant."

18. In Methods, TIGE2 should be TIEG2 (KLF11).

Response: Thank you for pointing out. We revised TIGE2 to TIEG2 (KLF11).

19. In Methods, please give more details on ChIP protocol: tissue shearing parameters, fragment length, use of input or IgG controls (see Comment #3)

Response: Thank you for pointing out. We added more detail about fragment length in method. **Page 15 L351-362.** Description of input were showed in result section. **Page 5 L106-109 and L112-114.**

"The mice were deeply anesthetized using a combination of three anesthetic agents: medetomidine hydrochloride (Domitol, Meiji Seika Pharma Co., Ltd., Tokyo, Japan, at 0.3 mg/kg), midazolam (Dormicum, Astellas Pharma Inc., Tokyo, Japan, at 4.0 mg/kg), and butorphanol (Vetorphale, Meiji Seika Pharma Co., Ltd., at 5.0 mg/kg), all administered intraperitoneally. They were then perfused transcardially with saline, followed by 1% paraformaldehyde at pH 7.4. The hippocampus was post-fixed for 10 minutes in 1% paraformaldehyde, after which 330 mM glycine was added. Subsequently, the chromatin was sheared into fragments of approximately 0.5–1 kb and immunoprecipitated using anti-HP1 γ antibody (1:250 for the hippocampal sample; Santa Cruz Biotechnology), anti-Histone H3 (trimethyl K9) antibody (1:100; Abcam, ab8898), anti-Histone H3 antibody (1:100; Abcam, ab10799), or TIEG2 (KLF11) antibody (1:100 Santa Cruz Biotechnology, sc-136101)."

"The results were expressed as a percentage of relative binding. This was calculated by comparing the ChIP assay signal (bound HP-1 γ) to the input sample signal (sample without anti-HP-1 γ)."

"The results were expressed as a percentage of relative binding. This was calculated by comparing the ChIP assay signal (bound H3K9me3) to the input sample signal (sample bound histone H3)."

20. In Results, precise that SEL is administered ip (although alternative strategies should probably replace this experiment, see Comment #1)

Response: According to reviewer's comment, we have included detailed information on the administration of SEL.

page 6, L134.

“(1 mg/kg, intraperitoneal administration for 3 times)”

21. Results text, Fig. 1I. Precise method used for dopamine levels quantification.

Response: According to reviewer’s comment, we added the method description in Fig.1I (newly Fig. 1J) result section.

Page 4, L79.

“After behavioral paradigm, we collected the hippocampus tissues from mice and measured dopamine level by ELISA.”

22. Results text, Fig. 2I. Precise this is MAOB protein expression, by contrast to mRNA levels above.

Response: We appreciate the Reviewer’s comment. We revised MAOB to MAOB protein expression.

Page 5, L91.

23. Introduction: “overexpression of these receptors in the dorsal raphe nucleus improves anxiety-like behavior”. Add reference.

Response: Thank you for pointing out. We added the reference.

#13 Shioda et al., J Neurosci 39 2019.

Reviewers' comments:

Reviewer #1 (Remarks to the Author):

I thank the authors for addressing my concerns.

1. In the abstract the sentence 'Here we found...' starting in line 18 and the next one 'CGRP administration' starting in line 21 still partially have the same content. Only the information on dopamine was omitted. Please revise also in terms of grammar!
2. Authors provide a nice explanation of their observed anxiogenic effects of selegiline. I think readers would benefit from this information as well and a discussion of the effects of selegiline should be included. Similarly, the information on the housekeeping genes should be provided to the reader.

Reviewer #2 (Remarks to the Author):

Authors have added several new experiments, new data analysis, and clarified figures, text and methods in a way that convincingly addresses my comments and questions. These additions and other revisions greatly improved the manuscript and its relevance. This reviewer particularly appreciates the addition of the MaoB siRNA experiment and of the KLF11-MaoB ChIP.

I have 2 final minor comments before recommending this manuscript for publication.

1. Please validate the in vivo efficiency of the MaoB siRNA knockdown (on MaoB RNA expression via qPCR at least).
2. t values and degrees of freedom should be reported in figure legends along with p values, as well as F values more consistently. Some are still missing.

We would like to thank you again for this opportunity to re-submit our manuscript. Our responses are listed below and the changes in the manuscript are highlighted with a blue background.

Reviewer #1 (Remarks to the Author):

1. In the abstract the sentence ‘Here we found...’ starting in line 18 and the next one ‘CGRP administration’ starting in line 21 still partially have the same content. Only the information on dopamine was omitted. Please revise also in terms of grammar!

Response: We appreciate for pointing out. We have revised the abstract as follows.

L18-L24

“We found that CGRP modulates anxiety behavior by epigenetically regulating the HP1 γ -KLF-11-MAOB pathway and depleting dopamine in the dorsal hippocampus. Intracerebroventricular administration of CGRP (0.5 nmol) elicited anxiety-like behaviors in open field, hole-board, and plus-maze tests. Additionally, we observed an increase in monoamine oxidase B (MAOB) levels and a concurrent decrease in dopamine levels in the dorsal hippocampus of mice following CGRP administration.”

2. Authors provide a nice explanation of their observed anxiogenic effects of selegiline. I think readers would benefit from this information as well and a discussion of the effects of selegiline should be included. Similarly, the information on the housekeeping genes should be provided to the reader.

Response: We appreciate reviewer’s comment. We revised the Discussion and Method sections as follows.

Page 10, L217-L226

“Selegiline, also known as L-deprenyl, is a medication primarily used to treat symptoms associated with Parkinson’s disease, as well as major depressive disorder in some cases. In our study, we observed that treatment selegiline alone exhibited an anxiogenic effect, evidenced by a reduction in the time spent in the open arms of the plus maze test and in the center area of the open filed test. One hypothesis is that intraperitoneal administration of selegiline might influence other brain regions. Additionally, administering selegiline three times before the behavioral testing could have amplified other effects. Notably, ‘anxiety’ is listed as a side effect in the selegiline drug package insert³⁵. While our current understanding is limited, it is important to note that selegiline, a MAOB inhibitor, appears to suppress CGRP-mediated anxiety response in mice.”

Page 15, L327-L328

“*Actin* was demonstrating the most stable cycle threshold values, was chosen as the house keeping gene.”

Reviewer #2 (Remarks to the Author):

1. Please validate the in vivo efficiency of the MaoB siRNA knockdown (on MaoB RNA expression via qPCR at least).

Response: According to your comment, we performed the western blot analysis both NTG and *MaoB*-siRNA knockdown (Fig. 5A). We also revised result Figure5, and Figure 5 legend as follows.

Page 8, L159-165

To confirm the impact of *MaoB* knockdown on MAOB expression in the mouse hippocampus, we quantified MAOB protein levels 4 days post-administration of *MaoB*-siRNA. This treatment resulted in a reduction of MAOB levels, as depicted in Figure 5A. To further investigate MAOB's role in CGRP-mediated angiogenesis, we administered either *MaoB*-siRNA or a non-targeting control (NTG) into the brains of mice on Day 1. Two days after the initial injection (Day 3), CGRP was administered directly to the hippocampus, followed by behavioral assessments the next day (Day 4) (Fig. 5B).

Page 24, L623-624

(A) MAOB protein expression 4 days following the administration of MAOB siRNA (n = 5 NTG, n = 6 siRNA, t = 2.203, df = 5.131, p = 0.0387)

2. t values and degrees of freedom should be reported in figure legends along with p values, as well as F values more consistently. Some are still missing.

Response: Thank you for pointing out. We carefully reviewed and included the t-values and df-values in the Figure Legends.